# A Bibliometric Analysis of QoR-15 Literature in Perioperative Recovery: Global Research Trends, Collaborations, and Citation Impact

**DOI:** 10.3390/healthcare13233051

**Published:** 2025-11-25

**Authors:** Abhijit Sukumaran Nair, Carel Jacobus de Wet, Sean Chetty, Sanjot Ninave

**Affiliations:** 1Department of Anaesthesiology, Datta Meghe Institute of Higher Education and Research (Deemed University), Sawangi Meghe, Wardha 442001, Maharashtra, India; 2Department of Anaesthesiology, Ibra Hospital, Ibra 413, Oman; 3Department of Anaesthesia, Livingstone Hospital Complex, Port Elizabeth 6020, South Africa; cjdewet7@gmail.com; 4Department of Anaesthesiology and Critical Care, Faculty of Medicine and Health Sciences, Stellenbosch University, Cape Town 7505, South Africa; seanchetty@sun.ac.za; 5Department od Anaesthesiology, Jawaharlal Nehru Medical College (JNMC), Sawangi-Meghe 442107, Maharashtra, India; drsusann02@rediffmail.com

**Keywords:** quality of recovery, patient-reported outcome measures, perioperative, bibliometric analysis

## Abstract

**Background/Objectives**: The 15-item Quality of Recovery questionnaire (QoR-15) is a widely adopted patient-reported outcome measure (PROM) in perioperative care. Although its validity and reliability have been confirmed across languages and surgical populations, no bibliometric analysis has systematically assessed its research impact, trends, and collaborative networks. **Methods**: We performed a bibliometric analysis of QoR-15 literature using the Scopus database (2013–June 2025). Eligible studies reporting on QoR-15 in perioperative or postoperative settings were included. Data were analyzed using VOS viewer (version 1.6.20) and CiteSpace (version 6.3) to evaluate co-authorship, keyword co-occurrence, citation patterns, bibliographic coupling, and co-citation networks. **Results**: Of 366 records screened, 341 articles were included. A total of 1901 authors, 999 organizations, and 43 countries contributed to the development of the QoR-15 literature. China produced the most publications (n = 164, 952 citations), whereas Australia generated fewer papers (n = 16) but with the highest citation impact (1479 citations). The most cited article was Myles et al. (year 2018, 219 citations). Co-occurrence mapping highlighted clusters in translation/validation studies, enhanced recovery pathways, and regional anesthesia applications. CiteSpace cluster analysis revealed emerging research in opioid-free anesthesia, pediatric adaptations, and frailty populations. **Conclusions**: This first bibliometric analysis of QoR-15 literature demonstrates rapid global adoption, with substantial contributions from Asia and high-impact outputs from Australia and Europe. Trends indicate increasing use of QoR-15 in regional and opioid-sparing strategies, emergency surgery, and pediatric care. These findings highlight QoR-15 as a robust, patient-centered endpoint and provide direction for future perioperative and pain research.

## 1. Introduction

Physical comfort, emotional stability, and functional independence are the essential aspects of a decent postoperative recovery. Traditional endpoints, such as morbidity, mortality, and length of hospital stay, do not adequately represent the subjective experiences of patients during the early stages of recovery. Clinical decision-making and research priorities are determined by patient-reported outcome measures (PROMs), offering clear insight into the quality of recovery from the patient’s perspective [1,2]. In recent years, patient-reported outcome measures (PROMs) have become central to perioperative quality improvement initiatives. One of the first elaborate PROMs in perioperative care was the Quality of Recovery-40 (QoR-40) questionnaire, which combined five domains into a total of 40 items: emotional state, physical comfort, psychological support, physical independence, and pain. Although the QoR-40 demonstrated strong validity and reliability, routine use in busy clinical settings and large trials was hampered by its length [3,4,5]. Generic and condition-specific tools such as the Short Form Health Survey (SF-36), EuroQOL 5 dimensions (EQ-5D, and Patient-Reported Outcomes Measurement Information System (PROMIS) instruments have complemented anesthesia-specific measures.

The Quality of Recovery-15 (QoR-15), a concise, one-page PROM created by choosing three representative items from each QoR-40 domain, was created by Stark et al. to mitigate these feasibility concerns [6]. Based on the results of a systematic review and meta-analysis, Kleif et al. concluded that the QoR15 tool fulfills all the requirements for measuring quality of patient recovery in the perioperative period [7]. To date, the QoR15 questionnaire has been successfully translated, culturally adapted, and validated in several languages worldwide, making it a globally applicable tool [8,9,10,11,12,13,14,15,16].

Bibliometric analysis provides a quantitative approach to mapping such research landscapes, thereby identifying evidence gaps that traditional narrative reviews may overlook. Despite the widespread adoption of QoR-15, to our knowledge, no prior bibliometric assessment has examined QoR-15 scholarship; existing reviews have been narrative or systematic in approach. Therefore, the objective of this review is to perform a comprehensive bibliometric analysis using validated tools to explore research hotspots, collaboration patterns, and citation networks related to the QoR-15 tool in perioperative care.

## 2. Materials and Methods

A comprehensive search of the Scopus database was executed on 10 July 2025. All records included were formally published by that date. The search strategy used was as follows: TITLE-ABS-KEY (“QoR-15” OR “QoR15” OR “quality of recovery-15” OR “quality of recovery 15”) AND TITLE-ABS-KEY (postoperative) AND PUBYEAR > 2012 AND PUBYEAR < 2026 AND (LIMIT-TO (LANGUAGE, ‘English’)). We restricted the term ‘postoperative’ after pilot searches, including ‘perioperative’ and ‘anesthesia’, produced large volumes of irrelevant results. Nevertheless, most included papers addressed the full perioperative continuum. Studies reporting on QoR-15 were defined as any document that administered the QoR-15 instrument either as a validation or translation study, used QoR-15 as an outcome measure in interventional or observational trials, or discussed QoR-15 substantively (systematic or narrative reviews mentioning or investigating QoR-15 used for the perioperative period). Duplicate records were removed in Scopus before export. The Scopus file was stored in comma-separated values (CSV) format. Two reviewers (ASN and JDW) independently screened all titles and abstracts; disagreements were resolved by consensus. Studies were eligible if QoR-15 was applied, validated, translated, or used as an outcome measure; mere mention without application led to exclusion. The final articles were processed for bibliometric analysis.

We used the VOS viewer (Version 1.6.20, Leiden University, The Netherlands) to perform bibliometric analysis across five categories: co-authorship, co-occurrence, citation, bibliographic coupling, and co-citation analysis. Countries, organizations, and authors were used as the units of analysis for the co-authorship analysis. The index, author, and all keywords were used as the units of analysis for co-occurrence. The units of analysis for citation and bibliographic coupling were documents, sources, authors, organizations, and nations. Citations for authors, sources, and references were the units of analysis for co-citation. Additionally, we used CiteSpace (version 6.3) to visualize and analyze temporal trends, keyword clusters, and research impact in the literature.

### Data Preparation and Analytical Parameters

Thresholds for inclusion were as follows: a minimum of 2 documents for authors and organisations, 5 occurrences for keywords, and 10 citations for cited references.

## 3. Results

The initial search of the Scopus database resulted in a total of 366 records. After screening all the titles and subsequent details, we excluded 25 titles that did not fulfil the inclusion criteria. Thus, a total of 341 titles fulfilled the criteria for bibliometric analysis. A PRISMA-style flow diagram is presented in Appendix A. The CSV file was then uploaded to the VOS viewer software for bibliometric analysis in various categories. In the figures, node size represents citation frequency, colour represents publication year, and line thickness represents link strength. For CiteSpace analysis, the CSV file was converted into a Web of Science-compatible format using a conversion tool to facilitate the analysis of bibliographic records.

### 3.1. Bibliometric Analysis Using VOS Viewer Software

#### 3.1.1. Co-Authorship Analysis

Analyzing co-authorship provides insights into collaborative dynamics among authors, universities, and countries, and also helps in diagnosing the patterns and connections between authors. Out of 1901 authors, 7 met the threshold (minimum two co-authored publications). Paul S. Miles had 7 documents with 1151 citations. Co-authorship between authors is summarized in Table 1A. Of the 999 organizations, 10 met the threshold. However, only three organizations were connected. The Department of Anesthesiology in Geneva, Switzerland, had 3 documents and 222 citations throughout. Co-authorship between organizations is summarized in Table 1B. Among 43 countries, 18 met the threshold. We created a network encompassing 18 countries and the top 10 countries. Although China contributed the highest volume of publications (n = 164), Australia produced fewer but higher-impact studies, reflected by the highest citations per article. Co-authorship between countries is summarized in Table 1C. Average citation impact varied markedly: Australia ≈ 92.4 citations per paper (1479/16), whereas China ≈ 5.8 citations per paper (952/164), highlighting differences in methodological impact and international visibility [17]). The relatively weak inter-institutional connectivity may reflect region-specific collaboration patterns and limited international joint funding.

#### 3.1.2. Co-Occurrence Analysis

Co-occurrence analysis examines the potential relationships between two bibliographic items that appear together, such as keywords or terms. This analysis facilitates mapping research topics and identifying thematic clusters, as well as understanding trends and research focus areas within a field. Out of 2555 keywords used, 462 met the threshold (≥5 occurrences). We created a network of 20 top keywords, as depicted in Figure 1. Of the 722 author keywords, 53 met the threshold. The network of 53 author keywords is depicted in Figure 2. We also generated a network using the top 20 author keywords. The most frequently used author keyword was ‘quality of recovery’ with an occurrence of 83 and a link strength of 97. Of the 2123 index keywords, 432 met the threshold. Figure 3 is a network created with 20 keywords. Clusters highlight thematic focus areas, including quality of recovery, enhanced recovery pathways, and regional anesthesia techniques.

#### 3.1.3. Citation Analysis

Citation analysis measures the impact and influence of published scientific works by examining citation frequency and patterns, and also facilitates the identification of significant and influential works within a field. It is also used to assess the popularity of research work and its impact on authors, articles, and journals.

Of the 341 documents, 206 met the threshold. We created a network of the top 10 documents (Table 2A). The document by Myles et al. in 2018 [18] had the maximum citations of 219. Of the 155 sources, 16 met the threshold. We created a network with all 16 sources (Figure 4). BJA had 18 documents and 964 citations for the documents. Of the 1901 authors, 7 met the threshold. We created a network with all 7 authors. The author, Paul S. Myles, had 1151 citations for 7 documents as an author (Table 2B). The prominence of Myles et al. reflects both foundational validation work and consensus definitions, establishing QoR-15 as the perioperative PROM of choice. Of the 999 organizations, 10 met the threshold. In this category, the Department of Anesthesiology at Geneva University Hospitals, Geneva, Switzerland had 222 citations, which was the maximum, for 3 documents (Table 2C). Of the 43 countries, 18 met the threshold. We created a network of the top 10 countries. China had 164 documents and 952 citations for the same (Figure 5). Differences in citation impact may relate to methodological rigor, multicentre trial visibility, and English-language accessibility of high-impact journals.

#### 3.1.4. Bibliographic Coupling

This analysis identifies similarity between documents according to common references cited by them. This is suggestive of a relationship between two different documents. The results of bibliographic analysis are important in identifying related research done historically and in related or recent documents. It also facilitates clustering documents or authors that feature similar references in their bibliography. Unlike co-citation, which reflects retrospective influence, bibliographic coupling identifies contemporary similarity based on shared references, highlighting emerging research fronts.

Out of 341 documents, 206 met the threshold. We created a network of 20 top documents. The document by Myles et al. [19] had the maximum citations (87) [Figure 6]. Out of 155 sources, 16 met the threshold. We created a network using all 16 sources. BJA had the maximum sources (18) and the highest citations (964) [Figure 7]. Out of the 1901 authors, 7 met the threshold. PS Myles had the maximum documents (7), with 1151 citations (Table 3A). Out of the 999 organizations, 10 met the threshold. We created a network with all 10 organizations (Table 3B). Cleveland Clinic, Ohio, United States, had the maximum documents (5) with 209 citations. However, the Department of Anesthesiology, Geneva University Hospital, Switzerland had the maximum citations (222) for 3 documents. Out of the 43 countries, 18 met the threshold. We selected the top 10 countries to create a network (Table 3C). China had 164 documents, which was the maximum, but had 952 citations. The maximum citations were from Australia, at 1479, for 16 documents.

#### 3.1.5. Co-Citation Analysis

Co-citation analysis generates a relationship between documents based on how often they are cited together by other documents. This analysis helps in clustering documents, authors, or journals by thematic similarity.

Of the 9221 references, 17 met the threshold. We created a network of the top 10 references. The 2013 document by Stark PA had the maximum citations, at 84 (Table 4A, Figure 8). Of the 2515 sources, 68 met the threshold. We created a network using the top 10 sources. BJA had the maximum citation among all sources, at 716 (Table 4B, Figure 9). Of the 22,123 authors, 113 met the threshold. We created a network using the top 10 authors. The author, PS Miles, had 584 citations, which was the maximum (Table 4C, Figure 10).

### 3.2. Bibliometric Analysis Using CiteSpace Software

Figure 11, which is generated using CiteSpace software, depicts cluster analysis of the research using QoR15 as the keyword. The cluster labelled #0, i.e., thoracic epidural (red), is the largest and most densely connected cluster, suggesting that thoracic epidural anesthesia is an important link between QoR15 outcomes and postoperative care. The cluster labelled #2, i.e., sequential analysis (yellow), suggests a methodological advance, emphasizing the use of sequential analysis and contemporary statistical frameworks in studies related to QoR15 studies, in the form of review articles, systematic reviews, and meta-analysis. Authors Myles PS, Kleif J, and Finnerty DT are the influential authors based on the repeatedly identified central nodes, suggesting significant influence and multiple contributions that generate evidence of the extensive use of QoR15 in various settings. The prominence of Professor Paul S. Myles within the network reflects his foundational contributions, from developing the QoR-40 (2000) and QoR-15 (2013) scales to defining patient-comfort endpoints through the StEP-COMPAC initiative (2018) [18]. These works collectively shaped psychometric standards and outcome reporting in perioperative medicine. Highly cited validation studies by Kleif et al. (2018) [7] and Myles et al. (2022) [19] further consolidated the tool’s methodological credibility. High modularity (Q = 0.829) and weighted mean silhouette (S = 0.9736) scores represent a well-established, structured research field with distinct clusters and high reliability in the bibliometric classification.

Various temporal trends, key authors, and research connectivity related to the QoR15 tool are depicted in Figure 12, which is generated using the CiteSpace software. The largest cluster, labelled #0-erector spinae plane, suggests considerable research focus on regional anesthesia techniques and their role in enhancing postoperative recovery, as measured by QoR15. Other prominent clusters seen are #1-15-item quality and #2-subjective quality, indicating the use of the QoR15 tool for subjective patient-centered outcomes. The network links (edges) demonstrate high interconnectivity between clusters and major publications (for example, from subjective quality research to practical applications in nerve blocks or enhanced recovery after surgery), suggesting cross-referencing between publications and their interconnectivity. The authors Myles PS, Kleif J, Chazapis M, Finnerty DT are the most influential ones based on the multiple nodes, and also have several citations, suggesting their impact in strengthening the evidence. A modularity (Q) score of 0.829 and a silhouette (S) score of 0.974 indicate a well-structured and internally consistent network, suggesting distinct thematic clusters with minimal overlap [23].

Emerging clusters such as ‘opioid-free anaesthesia’ and ‘erector spinae plane’ blocks parallel global trends toward multimodal, opioid-sparing strategies encouraged by ERAS guidelines [24]. These approaches emphasise functional recovery and patient comfort, which are domains directly captured by the QoR-15.

## 4. Discussion

This bibliometric analysis of QoR-15-related literature since its inception in 2013 reveals that it has been psychometrically evaluated in various surgical populations across multiple countries and has been successfully utilized by translating and validating it in many global languages, thus attesting to its broad applicability.

This bibliometric analysis reveals three dominant clusters. First, cross-cultural translation and validation studies have demonstrated the utility of the QoR15 tool irrespective of the language and the country. This reflects the global generalizability and psychometric robustness of this tool. Second, methodological syntheses and the literature review have been successful in demonstrating that QoR-15 is a valid, reliable, and clinically acceptable tool and reflects postoperative recovery satisfactorily. Third, the literature and keyword search have successfully established QoR-15 as a global postoperative recovery monitoring and improvement tool [25]. The discrepancy between China’s high output and lower citation impact may reflect language barriers and regional citation patterns, whereas Australia’s smaller but highly cited output suggests leadership in methodological rigor and global influence.

Patient-centered outcomes in perioperative care are of utmost importance, and with the availability of a simple QoR15 tool, research interest in this essential aspect has gained momentum. A systematic review and meta-analysis of 26 studies and 22,847 patients across 16 countries, involving 15 different languages, demonstrated excellent discriminant validity and reliability, and good convergent validity [19]. The authors concluded the QoR15 tool as a valid, reliable, and patient-centred metric that is appropriate for use in surgical patients.

Recent research has demonstrated its importance in patients undergoing emergency surgeries, which was otherwise not explored adequately previously, considering the time constraints and already existing patient perceptions about an emergency intervention [26]. The QoR15 tool has also been successfully used in the pediatric surgical population using a pictorial adaptation of the tool [27]. Future work can establish efficacy across frailty sub-groups.

The significance of the various studies published describing QoR15 differs based on the type of article (translation or validation studies vs. interventional or studies involving synthesis of results of various studies) and also based on the surgical category (elective vs. emergency or inpatient vs. ambulatory). Future research should explore digital platforms for QoR-15 administration, establishing the efficacy of the QoR-15 tool across various subpopulations, like patients with documented frailty, and various onco-surgeries. Development of normative datasets across age, sex, and surgical categories, and establishment of clinically meaningful score thresholds, will further enhance interpretability. Studies could integrate QoR-15 with longer-term PROMs and standardize the timing of assessment of the scores to optimize generalizability and comparability. In addition, QoR-15’s emphasis on pain, comfort, and independence aligns with growing emphasis on patient-centered pain outcomes, suggesting potential wider adoption in chronic and acute pain research.

The visualization using CiteSpace highlights the evolution of QoR15 as a robust tool facilitating postoperative outcome research, demonstrating the use of QoR15 in various setups, with evidence in the form of research and review articles, thus providing a platform for extensive research in the field of perioperative care in the future.

There are multiple limitations to this bibliometric analysis. Only the Scopus database was analysed owing to software limitations. Although Scopus offers broad coverage, it tends to over-represent English-language and high-impact journals, potentially under-capturing regional or non-indexed literature [17]. Inclusion of Web of Science, PubMed, Lens, or Dimensions databases may have broadened coverage. Future bibliometric analyses should triangulate multiple databases. Unfortunately, the software only allows analysis using a single database at a time. If we analyze using records gathered from several databases, the number of figures and tables will increase. Therefore, some articles might have been omitted in the analysis. Those published after the date of our data collection must have been excluded because new articles were constantly being added to the literature. The exclusion of non-English and grey literature may bias findings toward English-speaking and indexed journals.

While the QoR-15 demonstrates broad validity and feasibility, certain limitations warrant attention. Ceiling effects may occur in ambulatory populations with excellent recovery; conversely, frail or oncologic patients may require complementary measures such as QoR-40 or SF-36 to capture nuanced recovery trajectories. Despite widespread clinical use, integration into health-policy instruments such as national quality audits or ERAS dashboards remains limited. Explicit linkage of QoR-15 scoring to discharge planning and quality-improvement initiatives could enhance its translational value.

## 5. Conclusions

This bibliometric and scientometric analysis provides a comprehensive overview of the global research landscape surrounding the QoR-15 tool. QoR-15 has evolved from validation studies to a globally adopted, psychometrically robust endpoint. Bibliometric trends highlight increasing focus on regional anesthesia, opioid-sparing strategies, and frailty research, underscoring its future role in perioperative pain and recovery science. Future research should focus on establishing standardized postoperative time points for QoR-15 administration, longitudinal use of QoR-15 beyond the immediate postoperative period, and expanding its use in underrepresented populations, such as frail, oncologic, or low-literacy patients.

## Figures and Tables

**Figure 1 healthcare-13-03051-f001:**
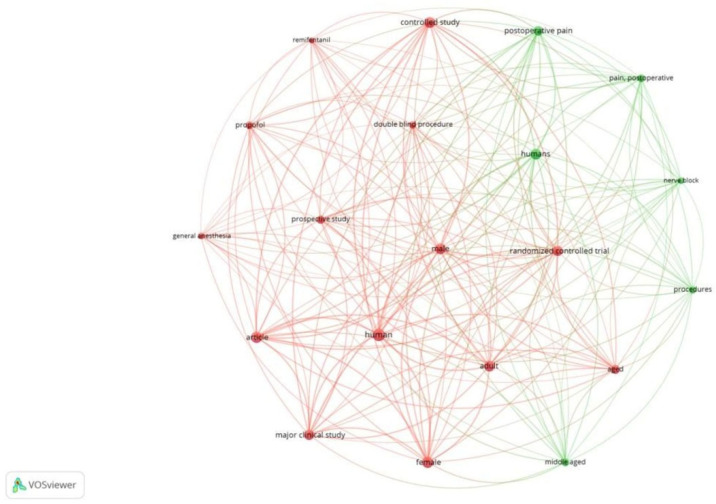
Co-occurrence and all keywords [Node size = citation frequency; color = publication year (warm = recent); line thickness = link strength].

**Figure 2 healthcare-13-03051-f002:**
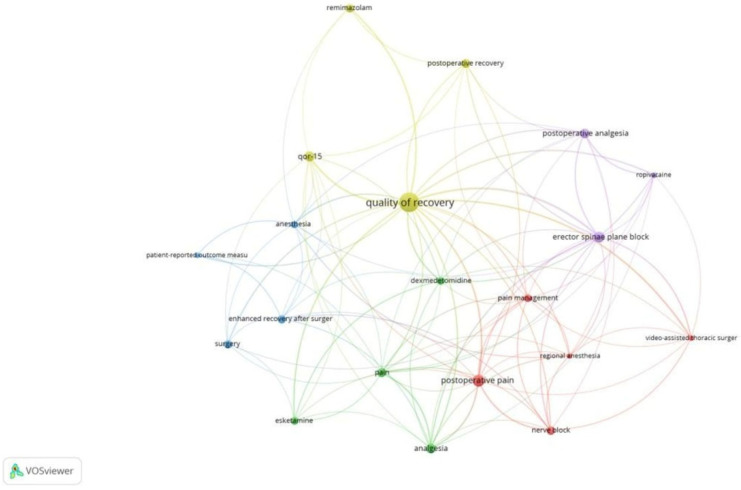
Co-occurrence and author keywords [Node size = citation frequency; color = publication year (warm = recent); line thickness = link strength].

**Figure 3 healthcare-13-03051-f003:**
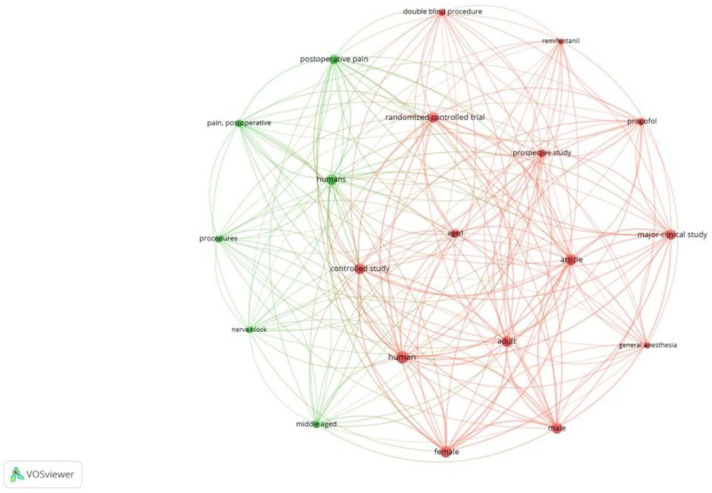
Co-occurrence and index keywords [Node size = citation frequency; color = publication year (warm = recent); line thickness = link strength].

**Figure 4 healthcare-13-03051-f004:**
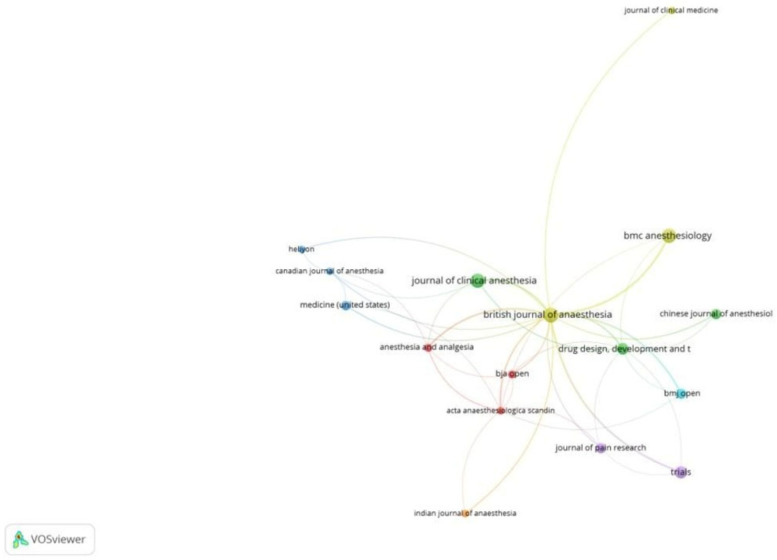
Citation analysis and sources [Node size = citation frequency; color = publication year (warm = recent); line thickness = link strength].

**Figure 5 healthcare-13-03051-f005:**
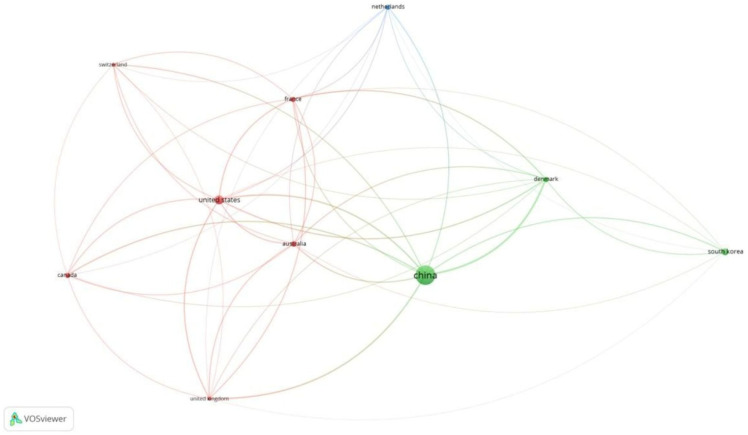
Citation analysis and countries [Node size = citation frequency; color = publication year (warm = recent); line thickness = link strength].

**Figure 6 healthcare-13-03051-f006:**
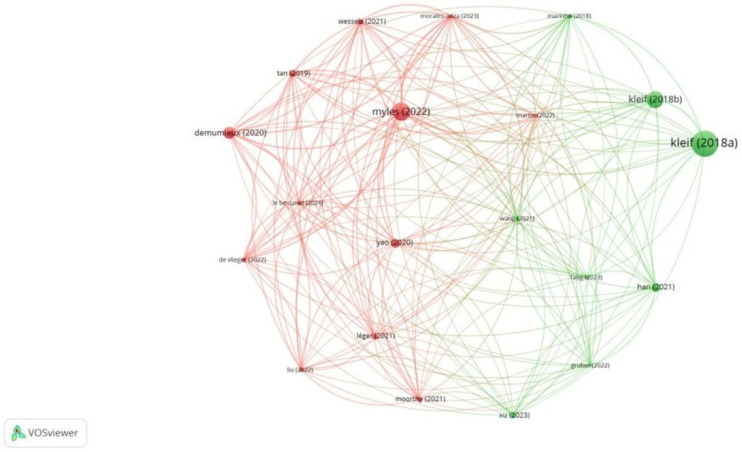
Bibliographic coupling and documents [Node size = citation frequency; color = publication year (warm = recent); line thickness = link strength].

**Figure 7 healthcare-13-03051-f007:**
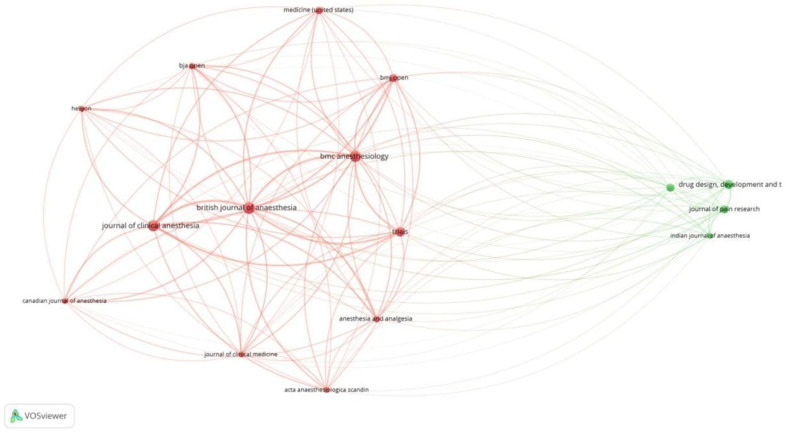
Bibliographic coupling and sources [Node size = citation frequency; color = publication year (warm = recent); line thickness = link strength].

**Figure 8 healthcare-13-03051-f008:**
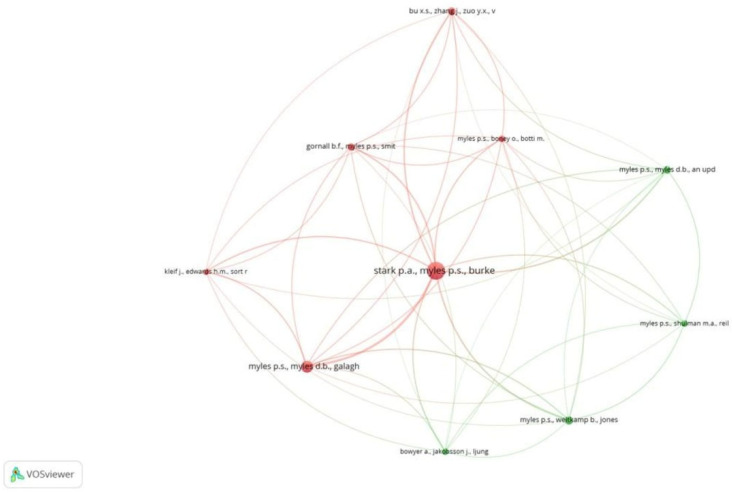
Co-citation analysis and references [Node size = citation frequency; color = publication year (warm = recent); line thickness = link strength].

**Figure 9 healthcare-13-03051-f009:**
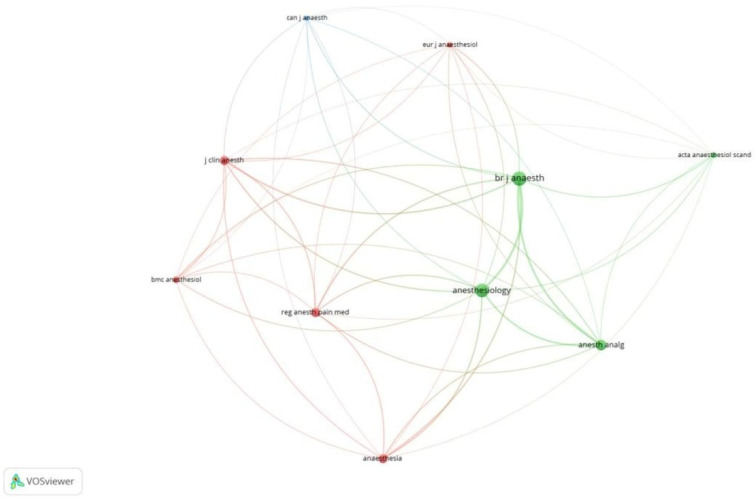
Co-citation analysis and sources [Node size = citation frequency; color = publication year (warm = recent); line thickness = link strength].

**Figure 10 healthcare-13-03051-f010:**
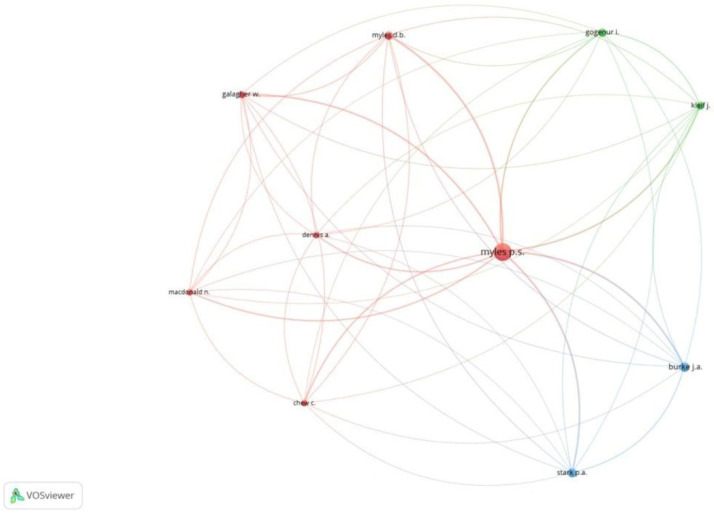
Co-citation analysis and authors [Node size = citation frequency; color = publication year (warm = recent); line thickness = link strength].

**Figure 11 healthcare-13-03051-f011:**
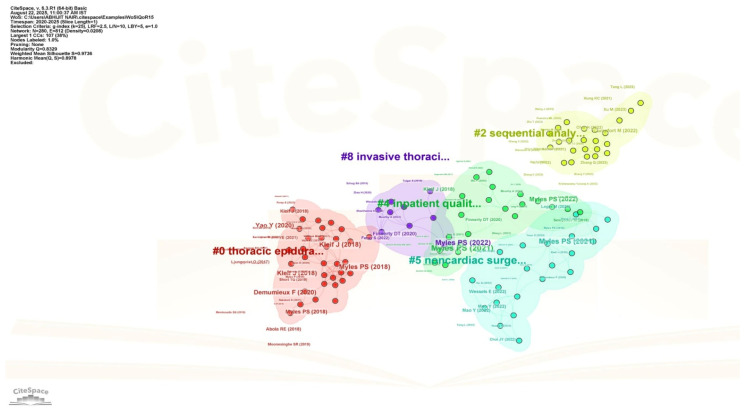
Cluster analysis of the research field centered on “QoR15” (Quality of Recovery-15) and “postoperative” outcomes [Node size = citation frequency; color = publication year (warm = recent); line thickness = link strength].

**Figure 12 healthcare-13-03051-f012:**
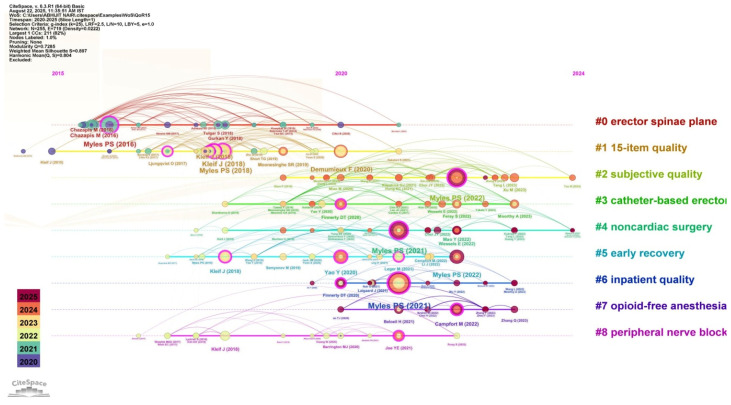
CiteSpace visualization summarizing temporal trends, key authors, research connectivity, and impact [Node size = citation frequency; color = publication year (warm = recent); line thickness = link strength].

**Table 1 healthcare-13-03051-t001:** Summary of co-authorship analysis (Threshold = minimum of 2 joint publications for authors or organisations; country networks include entities with ≥3 collaborative records).

A: Co-Authorship and Authors		
Author	Documents	Citations
Donal J Buggy	6	215
Ismail Gögenur	6	102
Mitsuru Ida	5	12
Masahiko Kawaguchi	5	12
Paul S Myles	7	1151
Ke Peng	5	10
Yusheng Yao	5	100
**B: Co-authorship and organizations**		
Organization	Documents	Citations
Department of Anaesthesiology, Geneva University Hospitals, Geneva, Switzerland	3	222
Department of Anesthesiology and Pain Medicine, University of California Davis Health, Sacramento, United States	3	10
Department of Anesthesiology, Chi Mei Medical Center, Tainan City, Taiwan	4	26
Department of Anesthesiology, Pain and Perioperative Medicine, the First Affiliated Hospital of Zhengzhou University, Henan, Zhengzhou, China	3	78
Department of Anesthesiology, The Affiliated Hospital of Xuzhou Medical University, Xuzhou, China	3	11
Department of Anesthesiology, West China Hospital, Sichuan University, Chengdu, China	3	26
Department of Psychiatry, Kaohsiung Chang Gung Memorial Hospital and Chang Gung University College of Medicine, Kaohsiung City, Taiwan	3	12
Division of Anaesthesiology, Mater Misericordiae University Hospital, Dublin, Ireland	3	196
Outcomes Research, Cleveland Clinic, Cleveland, United States	5	209
School of Medicine, University College, Dublin, Ireland	3	55
**C: Co-authorship and countries**		
Country	Documents	Citations
Australia	16	1479
Belgium	6	208
Canada	12	328
China	164	952
Denmark	13	613
Ireland	10	259
The Netherlands	12	331
Switzerland	6	251
United Kingdom	7	376
United States	35	743

**Table 2 healthcare-13-03051-t002:** Summary of citation analysis.

A: Citation Analysis and Documents		
Document	Year	Citations
Myles	2022	87
Tan	2019	13
Myles	2018	219
Bu	2016	99
Kleif	2018a	181
Demumieux	2020	37
Kleif	2018b	73
Kleif	2015	73
Yoon	2020	42
Chazapis	2016	118
**B: Citation analysis and authors**		
Author	Documents	Citations
Donal J Buggy	6	215
Ismail Gögenur	6	102
Mitsuru Ida	5	12
Masahiko Kawaguchi	5	12
Paul S. Myles	7	1151
Ke Peng	5	10
Yusheng Yao	5	100
**C: Citation and organization (same as bibliographic coupling)**		
Organization	Documents	Citations
Department of Anaesthesiology, Geneva University Hospitals, Geneva, Switzerland	3	222
Department of Anesthesiology and Pain Medicine, University of California, Davis Health, Sacramento, United States	3	10
Department of Anesthesiology, Chi Mei Medical Center, Tainan City, Taiwan	4	26
Department of Anesthesiology, Pain and Perioperative Medicine, The First Affiliated Hospital of Zhengzhou University, Henan, Zhengzhou, China	3	78
Department of Anesthesiology, the affiliated hospital of Xuzhou Medical University, Xuzhou, China	3	11
Department of Anesthesiology, West China Hospital, Sichuan University, Chengdu, China	3	26
Department of Psychiatry, Kaohsiung Chang Gung Memorial Hospital and Chang Gung University College of Medicine, Kaohsiung City, Taiwan	3	12
Division of Anaesthesiology, Mater Misericordiae University Hospital, Dublin, Ireland	3	196
Outcomes Research, Cleveland Clinic, Cleveland, United States	5	209
School of Medicine, University College, Dublin, Ireland	3	55

**Table 3 healthcare-13-03051-t003:** Summary of bibliographic coupling.

A: Bibliographic Coupling and Authors		
Author	Documents	Citations
Donal J Buggy	6	215
Ismail Gögenur	6	102
Mitsuru Ida	5	12
Masahiko Kawaguchi	5	12
Paul S Myles	7	1151
Ke Peng	5	10
Yusheng Yao	5	100
**B: Bibliographic coupling and organizations**		
Organization	Documents	Citations
Department of Anaesthesiology, Geneva University Hospitals, Geneva, Switzerland	3	222
Department of Anesthesiology and Pain Medicine, University of California, Davis Health, Sacramento, United States	3	10
Department of Anesthesiology, Chi Mei Medical Center, Tainan City, Taiwan	4	26
Department of Anesthesiology, Pain and Perioperative Medicine, The First Affiliated Hospital of Zhengzhou University, Henan, Zhengzhou, China	3	78
Department of Anesthesiology, the affiliated hospital of Xuzhou Medical University, Xuzhou, China	3	11
Department of Anesthesiology, West China Hospital, Sichuan University, Chengdu, China	3	26
Department of Psychiatry, Kaohsiung Chang Gung Memorial Hospital and Chang Gung University College of Medicine, Kaohsiung City, Taiwan	3	12
Division of Anaesthesiology, Mater Misericordiae University Hospital, Dublin, Ireland	3	196
Outcomes Research, Cleveland Clinic, Cleveland, United States	5	209
School of medicine, university college, Dublin, Ireland	3	55
**C: Bibliographic coupling and countries**		
Country	Documents	Citations
Australia	16	1479
Canada	12	328
China	164	952
Denmark	13	613
France	13	126
Ireland	10	259
The Netherlands	12	331
South Korea	22	289
United Kingdom	7	376
United States	35	743

**Table 4 healthcare-13-03051-t004:** Summary of co-citation analysis.

A: Co-Citation Analysis and Cited References	
Cited reference	Citations
Bowyer A et al. (2014) [20]	12
Bu XS et al. (2016) [10]	16
Gornall BF et al. (2013) [4]	15
Kleif et al. (2015) [9]	10
Myles PS et al. (2018) [18]	12
Myles PS et al. (2021) [21]	17
Myles PS et al. (2016) [22]	39
Myles PS et al. (2022) [19]	13
Myles PS et al. (2000) [3]	19
Stark PA et al. (2013) [6]	84
**B: Co-citation and sources**	
Source	Citations
Acta Anaesthesiologica Scandinavica	102
Anaesthesia	273
Anesthesia analgesia	403
Anesthesiology	678
BMC Anesthesiology	153
British Journal of Anaesthesia	716
Canadian Journal of Anaesthesiology	88
European Journal of Anaesthesiology	109
Journal of Clinical Anesthesia	283
Regional Anesthesia Pain Medicine	275
**C: Co-citation and authors**	
Author	Citations
JA Burke	167
C Chew	79
A Dennis	87
W Galagher	103
I Gogenur	138
J Kleif	122
N Macdonald	78
DB Myles	134
PS Myles	584
PA Stark	158

## Data Availability

No new data were created or analyzed in this study.

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
