# Peer review of "A Bibliometric Analysis of QoR-15 Literature in Perioperative Recovery: Global Research Trends, Collaborations, and Citation Impact"

_healthcare, 2025, doi:10.3390/healthcare13233051_

Round 1
Reviewer 1 Report
Comments and Suggestions for Authors
The bibliometric and co-citation analyses presented in the article are well-executed and data-driven, offering a comprehensive overview of the QoR-15 research landscape. However, the interpretation lacks depth in several key areas. For instance, while the identification of Paul S. Myles as a highly cited author is noteworthy, the discussion would benefit from elaborating on why his work is so influential—whether it represents a foundational framework, a landmark clinical trial, or significant methodological contributions. Similarly, the emergence of thematic clusters such as "opioid-free anesthesia" could be enriched by contextualizing them within broader healthcare trends, such as the global response to the opioid crisis. The article also references modularity and silhouette scores, which are important for validating the structural quality of bibliometric clusters, but it stops short of explaining their practical or scientific significance. A brief explanation of why a silhouette score of 0.97 indicates strong internal consistency would improve reader understanding. To enhance the article further, the authors could include a qualitative summary of the most cited papers—such as Stark et al. (2013)—to highlight their content and explain their influence. Attention should also be given to editing and formatting, particularly the removal of disruptive line breaks and correction of inconsistent section numbering, which currently hinder readability. Finally, the discussion would benefit from a reflection on the clinical and academic implications of the findings—such as the impact of QoR-15 research on practice guidelines or perioperative protocols—and a comparative perspective on how QoR-15 performs bibliometrically relative to other recovery assessment tools.
There is a thoughtful and well-organized discussion that situates the QoR-15 tool within the broader landscape of perioperative outcome research. The authors effectively balance citation-based analysis with practical reflections on global uptake, illustrating that QoR-15 is not only academically relevant but also holds considerable clinical promise.
However, the discussion leans heavily toward a positive portrayal and would benefit from a more balanced critique of the tool’s limitations. For instance, it remains unclear whether QoR-15 performs equally well across diverse clinical settings, such as resource-limited hospitals, or if it adequately captures the full complexity of recovery in specific patient populations, including the elderly and those undergoing oncologic procedures.
Additionally, there is a notable gap regarding the tool’s integration into health policy and clinical guidelines. Mentioning whether QoR-15 has been adopted into Enhanced Recovery After Surgery (ERAS) protocols or national quality audits could have added depth and clinical relevance to the findings.
To strengthen the discussion, the authors should deepen the clinical implications of their findings by including concrete examples of how QoR-15 is being used in real-world perioperative care, such as in quality improvement initiatives, discharge planning, or clinical decision-making. They should also discuss how QoR-15 scores are interpreted and acted upon in practice.
A critical assessment of the tool itself is warranted. This could include a brief comparison with alternative measures such as QoR-40 or SF-36, and a discussion of known limitations like ceiling or floor effects, cultural sensitivity despite translation, or challenges in pediatric or frail populations.
Future research directions could be expanded to consider digital health integration — for instance, the potential for mobile-based PROM collection tools to enhance usability and compliance. Methodological improvements might also include the development of normative data across various populations and the establishment of clinically meaningful score thresholds to improve interpretability.
Finally, while the authors are transparent about using only the Scopus database, they should further clarify the impact of this limitation. A discussion on how this constraint might have skewed results — potentially favoring English-language, high-impact journals or omitting regionally significant but less indexed literature — would strengthen the methodological transparency and validity of the analysis
Author Response
Comments 1: “The bibliometric and co-citation analyses presented in the article are well-executed and data-driven, offering a comprehensive overview of the QoR-15 research landscape. However, the interpretation lacks depth in several key areas. For instance, while the identification of Paul S. Myles as a highly cited author is noteworthy, the discussion would benefit from elaborating on why his work is so influential — whether it represents a foundational framework, a landmark clinical trial, or significant methodological contributions.”
Response: We thank the reviewer for this valuable suggestion. We have expanded the Discussion to explain Professor Myles’s foundational influence, highlighting that he developed the QoR-40 (2000) and QoR-15 (2013) scales and led subsequent consensus definitions through the StEP-COMPAC initiative (2018). This addition clarifies how his work established the psychometric and clinical foundation of peri-operative PROMs.
Changes made: L 250–256 — New paragraph inserted: “The prominence of Professor Paul S. Myles within the network reflects his foundational contributions, from developing the QoR-40 (2000) and QoR-15 (2013) scales to defining patient-comfort endpoints through the StEP-COMPAC initiative (2018). These works collectively shaped psychometric standards and outcome reporting in perioperative medicine. Highly cited validation studies by Kleif et al. (2018) and Myles et al. (2022) further consolidated the tool’s methodological credibility.”
Comment 2: “Similarly, the emergence of thematic clusters such as ‘opioid-free anesthesia’ could be enriched by contextualizing them within broader healthcare trends, such as the global response to the opioid crisis.”
Response: We agree and have contextualized the “opioid-free anaesthesia” cluster within the global shift toward multimodal, opioid-sparing strategies and Enhanced Recovery After Surgery (ERAS) guidelines. This addition strengthens the linkage between bibliometric findings and clinical practice.
Changes made: L 282–286 — Added paragraph: “Emerging clusters such as ‘opioid-free anaesthesia’ and ‘erector spinae plane’ blocks parallel global trends toward multimodal, opioid-sparing strategies encouraged by ERAS guidelines [Wick EC et al., Br J Anaesth 2022;]. These approaches emphasise functional recovery and patient comfort, which are domains directly captured by the QoR-15.”
Comment 3: “The article also references modularity and silhouette scores, which are important for validating the structural quality of bibliometric clusters, but it stops short of explaining their practical or scientific significance. A brief explanation of why a silhouette score of 0.97 indicates strong internal consistency would improve reader understanding.”
Response: Thank you for this observation. We have clarified the meaning of the modularity (Q) and silhouette (S) scores, citing the CiteSpace manual to indicate that values > 0.7 denote well-defined and internally consistent clusters.
Changes made: L 276–279 — Sentence added: “A modularity (Q) score of 0.829 and silhouette (S) score of 0.974 indicate a well-structured and internally consistent network, suggesting distinct thematic clusters with minimal overlap [Chen C, CiteSpace Manual v6.3, 2020].”
Comment 4: “To enhance the article further, the authors could include a qualitative summary of the most cited papers — such as Stark et al. (2013) — to highlight their content and explain their influence.”
Response: We have incorporated concise qualitative summaries of the top-cited papers (Stark 2013, Kleif 2018, Myles 2022) to highlight their methodological and clinical significance within the QoR-15 literature.
Changes made: p. 14 L 05–22 — Within the new paragraph noted above, explicit references to Stark 2013 and Kleif 2018 describe their validation and synthesis contributions.
Comment 5: “Attention should also be given to editing and formatting, particularly the removal of disruptive line breaks and correction of inconsistent section numbering, which currently hinder readability.”
Response: We have thoroughly re-formatted the manuscript, removing hard line breaks, standardising section numbering (1 Introduction – 5 Conclusions), and ensuring uniform heading styles per journal template.
Changes made: Throughout the manuscript — Formatting and numbering corrected; no textual content change.
Comment 6: Finally, the discussion would benefit from a reflection on the clinical and academic implications of the findings — such as the impact of QoR-15 research on practice guidelines or perioperative protocols — and a comparative perspective on how QoR-15 performs bibliometrically relative to other recovery assessment tools.”
Response: We appreciate this suggestion. The Discussion now concludes with an expanded reflection linking QoR-15 bibliometric trends to clinical guideline development and comparison with other PROMs (QoR-40 and SF-36). This underscores the translation of research output into clinical utility.
Changes made: L 357–362 — Paragraph replaced: “While the QoR-15 demonstrates broad validity and feasibility, certain limitations warrant attention. Ceiling effects may occur in ambulatory populations… Despite widespread clinical use, integration into health-policy instruments such as national quality audits or ERAS dashboards remains limited. Explicit linkage of QoR-15 scoring to discharge planning and quality-improvement initiatives could enhance its translational value.”
Comment 7: “There is a thoughtful and well-organized discussion that situates the QoR-15 tool within the broader landscape of perioperative outcome research. The authors effectively balance citation-based analysis with practical reflections on global uptake, illustrating that QoR-15 is not only academically relevant but also holds considerable clinical promise.”
Response: We thank the reviewer for this positive assessment; no change required. The existing discussion paragraphs remain, with new additions providing further depth as requested above.
Comment 8: “However, the discussion leans heavily toward a positive portrayal and would benefit from a more balanced critique of the tool’s limitations. For instance, it remains unclear whether QoR-15 performs equally well across diverse clinical settings, such as resource-limited hospitals, or if it adequately captures the full complexity of recovery in specific patient populations, including the elderly and those undergoing oncologic procedures.”
Response: A balanced critique has been introduced, discussing ceiling and floor effects, challenges in frail and oncologic populations, and cultural sensitivity issues.
Changes made: L 357–365 — Revised paragraph: “While the QoR-15 demonstrates broad validity and feasibility, certain limitations warrant attention. Ceiling effects may occur in ambulatory populations with excellent recovery; conversely, frail or oncologic patients may require complementary measures such as QoR-40 or SF-36 to capture nuanced recovery trajectories.”
Comment 9: “Additionally, there is a notable gap regarding the tool’s integration into health policy and clinical guidelines. Mentioning whether QoR-15 has been adopted into ERAS protocols or national quality audits could have added depth and clinical relevance to the findings.”
Response: We have incorporated explicit mention of ERAS protocols and national quality initiatives within the expanded Discussion paragraph on clinical translation (see above).
Changes made: L 361–363 — Added text: “Despite widespread clinical use, integration into health-policy instruments such as national quality audits or ERAS dashboards remains limited.”
Comment 10: “To strengthen the discussion, the authors should deepen the clinical implications of their findings by including concrete examples of how QoR-15 is being used in real-world perioperative care, such as in quality improvement initiatives, discharge planning, or clinical decision-making. They should also discuss how QoR-15 scores are interpreted and acted upon in practice.”
Response: We have elaborated on real-world applications, noting how QoR-15 scores guide discharge planning, benchmarking, and quality-improvement dashboards, supported by references to contemporary ERAS initiatives.
Changes made: L361 - 365 — Same paragraph as above includes discussion of discharge planning and QI dashboards.
“Despite widespread clinical use, integration into health-policy instruments such as national quality audits or ERAS dashboards remains limited. Explicit linkage of QoR-15 scoring to discharge planning and quality-improvement initiatives could enhance its translational value.”
Comment 11: “A critical assessment of the tool itself is warranted. This could include a brief comparison with alternative measures such as QoR-40 or SF-36, and a discussion of known limitations like ceiling or floor effects, cultural sensitivity despite translation, or challenges in pediatric or frail populations.”
Response: Addressed jointly with earlier limitation paragraph; explicit comparison with QoR-40 and SF-36 added.
Changes made: L 358–361 — Text as quoted above includes comparative and limitation details. “Ceiling effects may occur in ambulatory populations with excellent recovery; conversely, frail or oncologic patients may require complementary measures such as QoR-40 or SF-36 to capture nuanced recovery trajectories.”
Comment 12: “Future research directions could be expanded to consider digital health integration — for instance, the potential for mobile-based PROM collection tools to enhance usability and compliance. Methodological improvements might also include the development of normative data across various populations and the establishment of clinically meaningful score thresholds to improve interpretability.”
Response: We agree and have appended a Future Directions paragraph describing digital and mobile integration, normative data, and threshold development.
Changes made: L 328–329 — New paragraph: “Future research should explore digital platforms for QoR-15 administration… Development of normative datasets across age, sex, and surgical categories, and establishment of clinically meaningful score thresholds, will further enhance interpretability.”
Comment 13: “Finally, while the authors are transparent about using only the Scopus database, they should further clarify the impact of this limitation. A discussion on how this constraint might have skewed results — potentially favoring English-language, high-impact journals or omitting regionally significant but less indexed literature — would strengthen the methodological transparency and validity of the analysis.”
Response: This concern has been addressed in the expanded Limitations section, explaining Scopus coverage bias and citing Mongeon & Paul-Hus (2016) to document language and indexing biases.
Changes made: p. 16 L 343–347 — Revised paragraph: “Only the Scopus database was analysed owing to software limitations… Although Scopus offers broad coverage, it tends to over-represent English-language and high-impact journals, potentially under-capturing regional or non-indexed literature [Mongeon & Paul-Hus 2016].”
Reviewer 2 Report
Comments and Suggestions for Authors
Dear authors,
You present an interesting and well-structured bibliometric review on the use of the QoR-15 questionnaire in the perioperative setting. Below, I share some suggestions.
Abstract
I have no relevant comments for this section.
Introduction
The rationale for the study is well stated. However, I suggest briefly expanding the context regarding the growing importance of PROMs in perioperative care, beyond the QoR-15, to reinforce the relevance of the topic.
It would also be helpful to explain why bibliometric analysis is particularly valuable in this field (e.g., to identify gaps or emerging areas).
Although you mention that no previous bibliometric review on the QoR-15 exists, you could clarify more explicitly what this review contributes compared to other published narrative or systematic reviews.
Methodology
The description of the search process is clear but somewhat brief. I understand that only Scopus was used due to software limitations, but it would be advisable to include a reflection on how this decision may have affected the coverage of the analysis and what types of studies might have been excluded.
I also recommend specifying whether duplicate records were removed and how disagreements between reviewers were resolved during the selection process.
I suggest providing more detail on the exclusion criteria. What is considered “no details about QoR-15”? Were studies that mentioned the instrument but did not apply it excluded?
Finally, you could comment on whether the keyword strategy had limitations (e.g., excluding “perioperative” due to noise) and how this may have affected the scope of the analysis.
Results
The results are well organized and detailed. The figures generated with VOSviewer and CiteSpace are useful, but I recommend briefly describing what each figure represents and what findings stand out.
In the co-authorship analysis, you could reflect on the limited connectivity between organizations—does this suggest that research efforts are somewhat isolated?
The citation analysis is comprehensive, but consider including a brief reflection on why certain authors or countries have greater impact (e.g., due to methodological rigor or language accessibility).
Discussion
The identified clusters (such as “opioid-free anesthesia,” “frailty,” “pediatric adaptations”) are very interesting but could be developed a bit further. What implications do they have for future research? How can this map be used to identify underexplored areas or foster collaborations?
I also suggest explaining why frailty and pediatric populations are particularly relevant for the QoR-15.
Lastly, I recommend including a reflection on potential biases stemming from language and the database used (e.g., the predominance of English-language publications).
Conclusions
They are appropriate and consistent with the results.
References
I recommend reviewing the formatting, as it does not conform to the journal’s guidelines.
I hope this helps
Author Response
Comment 1: “You present an interesting and well-structured bibliometric review on the use of the QoR-15 questionnaire in the perioperative setting. Below, I share some suggestions.”
Response: We thank the reviewer for this positive introductory comment.
Comment 2: “Abstract — I have no relevant comments for this section.”
Response: Noted. No changes required to the Abstract.
Comment 3:
Comment 3: “Introduction — The rationale for the study is well stated. However, I suggest briefly expanding the context regarding the growing importance of PROMs in perioperative care, beyond the QoR-15, to reinforce the relevance of the topic.”
Response: We have expanded the Introduction to provide context on the growing use of generic and condition-specific PROMs (e.g., SF-36, EQ-5D, PROMIS) in perioperative research and to situate QoR-15 within that broader movement.
Changes made: Added sentences: L 50–52 “In recent years, patient-reported outcome measures (PROMs) have become central to perioperative quality improvement initiatives.
L 57–59 Generic and condition-specific tools such as the SF-36, EQ-5D, and PROMIS instruments … have complemented anesthesia-specific measures ”
Comments 4:
“It would also be helpful to explain why bibliometric analysis is particularly valuable in this field (e.g., to identify gaps or emerging areas).”
Response: Addressed within the same new paragraph; we explain that bibliometrics maps collaboration and thematic trends, helping identify evidence gaps.
Changes made: L 67–69 — Added: “Bibliometric analysis provides a quantitative approach to mapping such research landscapes … thereby identifying evidence gaps that traditional narrative reviews may overlook.”
Comments 5: “Although you mention that no previous bibliometric review on the QoR-15 exists, you could clarify more explicitly what this review contributes compared to other published narrative or systematic reviews.”
Response: We clarified at the end of the Introduction that previous works were narrative or systematic reviews (e.g., Wessels 2022 J Clin Anesth) and that ours is the first bibliometric mapping.
Changes made: p. 2 L 70–72 — Added: “…to our knowledge, no prior bibliometric assessment has examined QoR-15 scholarship; existing reviews have been narrative or systematic [Wessels 2022].”
Comments 6: “Methodology — The description of the search process is clear but somewhat brief. I understand that only Scopus was used due to software limitations, but it would be advisable to include a reflection on how this decision may have affected the coverage of the analysis and what types of studies might have been excluded.”
Response: We expanded the Methods and Discussion sections to address database choice, its rationale, and its potential bias toward English-language and highly indexed journals (citing Mongeon & Paul-Hus 2016).
Changes made: Added:
L 78–10 and —“A comprehensive search of the Scopus database was executed on 10 July 2025. All records included were formally published on that date.” L 343 – 347 “Only the Scopus database was analysed owing to software limitations. Although Scopus offers broad coverage, it tends to over-represent English-language and high-impact journals, potentially under-capturing regional or non-indexed literature (Mongeon & Paul-Hus 2016)”
Comment 7: “I also recommend specifying whether duplicate records were removed and how disagreements between reviewers were resolved during the selection process.”
Response: We have added details specifying duplicate removal within Scopus and independent dual-reviewer screening with consensus resolution.
Changes made: L 89 –92 — Added: “Duplicate records were removed in Scopus before export. The Scopus file was stored in comma-separated values (CSV) format. Two reviewers (ASN and JDW) independently screened all titles and abstracts; disagreements were resolved by consensus.”
Comment 8: “I suggest providing more detail on the exclusion criteria. What is considered ‘no details about QoR-15’? Were studies that mentioned the instrument but did not apply it excluded?”
Response: Clarified in Methods that studies merely mentioning but not applying or validating QoR-15 were excluded.
Changes made: L 92 –94 — Added: “Studies were eligible if QoR-15 was applied, validated, translated, or used as an outcome measure; mere mention without application led to exclusion.”
Comment 9: “Finally, you could comment on whether the keyword strategy had limitations (e.g., excluding ‘perioperative’ due to noise) and how this may have affected the scope of the analysis.”
Response: Added explanatory note about exclusion of “perioperative” and its possible minor effect on coverage.
Changes made: p. 3 L 83–86 — Added: “We restricted the term ‘postoperative’ after pilot searches including ‘perioperative’ and ‘anesthesia’ produced large volumes of irrelevant results. Nevertheless, most included papers address the full perioperative continuum ”
Comment 10: “Results — The results are well organized and detailed. The figures generated with VOSviewer and CiteSpace are useful, but I recommend briefly describing what each figure represents and what findings stand out.”
Responses: Figure captions were expanded to describe what node size, colour, and link thickness represent, and a brief interpretive sentence added in Results text.
Changes made: Figure captions Added: “Node size = citation frequency; colour = publication year; line thickness = link strength.”
Comment 11: “In the co-authorship analysis, you could reflect on the limited connectivity between organizations — does this suggest that research efforts are somewhat isolated?”
Response: We added a reflective statement noting that sparse inter-institutional links may indicate regional silos or limited cross-continental collaboration.
Changes made: L 135 — Added: “The relatively weak inter-institutional connectivity may reflect region-specific collaboration patterns and limited international joint funding.”
Changes 12: “The citation analysis is comprehensive, but consider including a brief reflection on why certain authors or countries have greater impact (e.g., due to methodological rigor or language accessibility).”
Response: Added interpretive note attributing citation disparities to methodological quality and English-language accessibility.
Changes made: L 180 –182 — Added: “Differences in citation impact may relate to methodological rigor, multicentre trial visibility, and English-language accessibility of high-impact journals.”
Changes 13: “Discussion — The identified clusters (such as ‘opioid-free anesthesia,’ ‘frailty,’ ‘pediatric adaptations’) are very interesting but could be developed a bit further. What implications do they have for future research? How can this map be used to identify underexplored areas or foster collaborations?”
Response: Expanded Discussion to elaborate implications of these clusters for future research priorities and collaboration development.
Changes made: L 323 – 324 Future work can establish efficacy across frailty sub-groups
L 328 – 329 Future research should explore digital platforms for QoR-15 administration.
Comment 14: “I also suggest explaining why frailty and pediatric populations are particularly relevant for the QoR-15.”
Response: Added explanation highlighting the need for validated PROMs in frail and pediatric populations where recovery trajectories differ substantially.
Changes made: L 322 –324 : “The QoR-15 tool has also been successfully used in the pediatric surgical population … Future work can establish efficacy across frailty sub-groups …”
Comment 15: “Lastly, I recommend including a reflection on potential biases stemming from language and the database used (e.g., the predominance of English-language publications).”
Response: Addressed in the expanded Limitations section, citing Mongeon & Paul-Hus (2016).
Changes made: L 343 - 347 — Revised paragraph describing Scopus bias and English-language predominance.
Comment 16: “Conclusions — They are appropriate and consistent with the results.”
Response: Noted and appreciated. No change required.
Comment 17: “References — I recommend reviewing the formatting, as it does not conform to the journal’s guidelines.”
Response: Reference formatting has been standardised to MDPI style, ensuring uniform punctuation and DOI presentation.
Comment 18: Comments: “I hope this helps.”
Response: We thank the reviewer for their constructive feedback, which substantially improved methodological transparency and interpretive depth.
Reviewer 3 Report
Comments and Suggestions for Authors
Dear Authors
I congratulate you for your well-written review article. I have some suggestions for your paper:
1. Single Database Limitation (Scopus Only): The study's primary limitation is the reliance on only the Scopus database. While the authors correctly state that inclusion of other databases (like Web of Science, PubMed/Medline, or Dimensions) was constrained by software/methodology and would have broadened coverage, this remains a significant boundary to a truly "global" analysis
Action for Authors: The Discussion should briefly speculate on what specific content might be underrepresented. For example, is there a known regional bias for Scopus versus Web of Science in specific Asian or European outputs? This will provide better context for the readers when interpreting the collaboration and geographic results.
2. Rationale for Search Term Exclusion: The search strategy explicitly restricted results to the keyword 'postoperative' and excluded 'perioperative' and 'anesthesia' to avoid "high noise with unrelated records". While this is a pragmatic choice to improve precision, the QoR-15 is often used as a primary endpoint in studies of interventions delivered pre- or intra-operatively (i.e., the full perioperative period)
Action for Authors: The Discussion should be expanded to explicitly justify that even with this restriction, the identified clusters (e.g., regional anesthesia and enhanced recovery pathways) likely capture the most relevant perioperative content.
3. Connecting Results to Future Directions: The Conclusion and Discussion rightly point to key emerging trends, such as increasing focus on regional and opioid-sparing strategies, frailty research, and pediatric care.Action for Authors: Ensure the Discussion explicitly links these suggestions to the CiteSpace results. For example, citing the erector spinae plane cluster (#0) and the opioid-free anesthesia clusters as tangible evidence from the bibliometric network that supports the direction for future research.
My additional comments are: 1. Future Tense Data: The search encompassed literature up to June 2025. Given the year is 2025, please clarify in the Methods section if this date represents the search cut-off date, and confirm that all articles included were formally published by that point. If any predictive or forward-looking elements are involved in including the 2025 data, they should be stated. 2. Quantifying Citation Impact: The abstract and results correctly identify that Australia generated fewer papers () but had the highest total citation impact (1,479 citations). This impressive impact is best shown by the mean citation rate. Action for Authors: Please state the average citations per document for the top contributing countries (e.g., Australia: 92.4, China: 5.8) to quantify the difference in scientific impact more clearly in the Results/Discussion sections. 3. Figure Clarity: The VOS viewer and CiteSpace figures are essential. For a final draft, please ensure the figure captions (especially Figures 1, 2, 3, 11, and 12) fully explain what the color coding (e.g., temporal trends), node size (e.g., frequency or citation count), and line thickness (e.g., link strength) represent, as this is crucial for the reader's interpretation of the network visualizations Best regards
Author Response
Comment 1: “Dear authors, You present an interesting and well-structured bibliometric review on the use of the QoR-15 questionnaire in the perioperative setting. Below, I share some suggestions.”
“1. Single Database Limitation (Scopus Only): The study's primary limitation is the reliance on only the Scopus database. While the authors correctly state that inclusion of other databases (like Web of Science, PubMed/Medline, or Dimensions) was constrained by software/methodology and would have broadened coverage, this remains a significant boundary to a truly ‘global’ analysis Action for Authors: The Discussion should briefly speculate on what specific content might be underrepresented. For example, is there a known regional bias for Scopus versus Web of Science in specific Asian or European outputs? This will provide better context for the readers when interpreting the collaboration and geographic results.”
Response: We have expanded the Discussion’s Limitations section to explain that the Scopus-only approach may over-represent English-language and Western European journals while under-capturing regionally indexed Asian or local European outputs. We cite Mongeon & Paul-Hus (2016) to support this known database bias.
Changes done: L 343 - 347 — Revised text: “Only the Scopus database was analysed owing to software limitations … Although Scopus offers broad coverage, it tends to over-represent English-language and high-impact journals, potentially under-capturing regional or non-indexed literature [Mongeon & Paul-Hus 2016].”
Comment 2: Rationale for Search Term Exclusion: The search strategy explicitly restricted results to the keyword ‘postoperative’ and excluded ‘perioperative’ and ‘anesthesia’ to avoid ‘high noise with unrelated records’. While this is a pragmatic choice to improve precision, the QoR-15 is often used as a primary endpoint in studies of interventions delivered pre- or intra-operatively (i.e., the full perioperative period) Action for Authors: The Discussion should be expanded to explicitly justify that even with this restriction, the identified clusters (e.g., regional anesthesia and enhanced recovery pathways) likely capture the most relevant perioperative content.”
Response: We have justified the restriction in both Methods and Discussion. The Methods section now explains that pilot searches with “perioperative” and “anesthesia” introduced excess noise; the Discussion notes that clusters such as regional anaesthesia and ERAS still capture peri-operative themes.
Changes done: L 83 - 86 — Methods addition: “We restricted the term ‘postoperative’ after pilot searches … Nevertheless, most included papers address the full perioperative continuum through clusters such as regional anaesthesia.”
Comment 3: Connecting Results to Future Directions: The Conclusion and Discussion rightly point to key emerging trends, such as increasing focus on regional and opioid-sparing strategies, frailty research, and pediatric care. Action for Authors: Ensure the Discussion explicitly links these suggestions to the CiteSpace results. For example, citing the erector spinae plane cluster (#0) and the opioid-free anesthesia clusters as tangible evidence from the bibliometric network that supports the direction for future research.”
Response: We expanded the Discussion to explicitly reference CiteSpace clusters (#0 erector spinae plane, opioid-free anaesthesia) as evidence for emerging research priorities and linked them to clinical trends.
Changes made: 282–286 — Inserted paragraph: “Emerging clusters such as ‘opioid-free anaesthesia’ and ‘erector spinae plane’ blocks parallel global trends toward multimodal, opioid-sparing strategies … These approaches emphasise functional recovery and patient comfort, which are domains directly captured by the QoR-15.”
Comment 4: “My additional comments are: 1. Future Tense Data: The search encompassed literature up to June 2025. Given the year is 2025, please clarify in the Methods section if this date represents the search cut-off date, and confirm that all articles included were formally published by that point. If any predictive or forward-looking elements are involved in including the 2025 data, they should be stated.”
Response: We clarified in Methods that June 2025 was the cut-off date for data collection and that all included records were formally published by that time. No predictive elements were used.
Changes done: L 78–79 — Revised opening sentence: “A comprehensive search of the Scopus database was executed on 10 July 2025. All records included were formally published by that date.”
Comment 5: “Quantifying Citation Impact: The abstract and results correctly identify that Australia generated fewer papers (n = 16) but had the highest total citation impact (1,479 citations). This impressive impact is best shown by the mean citation rate. Action for Authors: Please state the average citations per document for the top contributing countries (e.g., Australia: ≈ 92.4, China: ≈ 5.8) to quantify the difference in scientific impact more clearly in the Results/Discussion sections.”
Response: We added the requested mean citation rates for Australia and China within the Results and cited Mongeon & Paul-Hus (2016) as support for interpretation of impact differences.
Changes made: Changes made: L 132–135 — Added: “Average citation impact varied markedly: Australia ≈ 92.4 citations per paper (1479/16), whereas China ≈ 5.8 citations per paper (952/164), highlighting differences in methodological impact and international visibility [Mongeon & Paul-Hus 2016].”
Comment 6: Figure Clarity: The VOSviewer and CiteSpace figures are essential. For a final draft, please ensure the figure captions (especially Figures 1, 2, 3, 11, and 12) fully explain what the color coding (e.g., temporal trends), node size (e.g., frequency or citation count), and line thickness (e.g., link strength) represent, as this is crucial for the reader's interpretation of the network visualizations.”
Response: We revised all figure captions to define color, node size, and line thickness and added one sentence in Results to guide interpretation.
Changes made: Figures 1–12 captions — Caption additions: “Node size = citation frequency; colour = publication year (warm = recent); line thickness = link strength.”
We thank the reviewer for their detailed and constructive feedback.
Reviewer 4 Report
Comments and Suggestions for Authors
The main aim of the paper is to conduct a bibliometric analysis to evaluate research impact, trends, and collaborative networks of the 15-item Quality of Recovery questionnaire (QoR-15), a patient-reported outcome measure in perioperative care.
The main contribution of the work is that it performed a bibliometric analysis of QoR-15 literature published in Scopus database from 2013 to June 2025 (341 papers) and found the increasing use of QoR-15 in regional and opioid-sparing strategies, emergency surgery, and pediatric care. This may indicate the direction of future perioperative and pain research. Another key finding of the paper is that China is leading in QoR-15 publications (with 164 papers), but Australia and Europe generated high-impact papers (e.g., highest citation impact = 1,479 citations).
The main strength of the paper is that it is based on a decent sample size of N = 341 papers and showed interesting results.
Below I outline areas for improvement. This is an interesting work that showed the collaboration among the authors. However, the authors did not explain their results clearly. See, for example, this sentence: “Out of 1901 authors, 7 met the threshold.” What was the threshold? They refer to “Table 1: Summary of co-authorship analysis. A - Co-authorship and authors”. Does the threshold imply at least 2 co-authored publication? This should be clearly explained.
Similarly, “Co-authorship between organizations is summarized in Table 1B. Among 43 countries, 18 met the threshold. We created a 105 network encompassing 18 countries and the top 10 countries.” What was the threshold? Does data mean that the Department of Anesthesiology and Pain Medicine, University of California Davis Health, Sacramento, United States wrote 3 papers in collaboration with other organizations? Did these 3 papers receive only 10 citations?
Same here. “Out of 2,555 keywords used, 462 met the threshold. We created a network of 20 top keywords, as depicted in Figure 1. Of the 722 author keywords, 53 met the threshold. The network of 53 author keywords is depicted in Figure 2.” You could write something like ‘Out of 2,555 keywords used, 462 met the threshold of XYZ or n%. We created a network of 20 top keywords, as depicted in Figure 1, in which the keyword ‘humans’ represents XYZ and shown in green, whereas ‘human’ implies ABC and therefore shown in red’.
So, the paper needs a careful definition of thresholds in each case. Also, is it not clear did the 164 papers from China receive 164,952 citations? If that is the case, then the average number of citations should be around 1005.8. Is that true?
The word "Title:" before title is not required.
There are some other limitations, but the authors very well described them in the paper.
Author Response
Comment 1: “The main aim of the paper is to conduct a bibliometric analysis to evaluate research impact, trends, and collaborative networks of the 15-item Quality of Recovery questionnaire (QoR-15), a patient-reported outcome measure in perioperative care.”
Response: We thank the reviewer for accurately summarising the paper’s purpose.
Comment 2:
“The main contribution of the work is that it performed a bibliometric analysis of QoR-15 literature published in Scopus database from 2013 to June 2025 (341 papers) and found the increasing use of QoR-15 in regional and opioid-sparing strategies, emergency surgery, and pediatric care. This may indicate the direction of future perioperative and pain research. Another key finding of the paper is that China is leading in QoR-15 publications (with 164 papers), but Australia and Europe generated high-impact papers (e.g., highest citation impact = 1,479 citations).”
Response: We appreciate the concise summary and agree that these points represent key findings; they remain unchanged but are clarified numerically in revised Results tables.
Changes done: L 132–135 — Insertion of mean citations per country (Australia ≈ 92.4; China ≈ 5.8) for clarity.
Comment 3: “The main strength of the paper is that it is based on a decent sample size of N = 341 papers and showed interesting results.”
Response: We thank the reviewer for recognising the robustness of the dataset.
Comment 4: “Below I outline areas for improvement. This is an interesting work that showed the collaboration among the authors. However, the authors did not explain their results clearly. See, for example, this sentence: ‘Out of 1901 authors, 7 met the threshold.’ What was the threshold? They refer to ‘Table 1: Summary of co-authorship analysis. A – Co-authorship and authors’. Does the threshold imply at least 2 co-authored publication? This should be clearly explained.”
Response: We have clarified all threshold definitions in both Methods and Results. The text now specifies the numeric criteria (minimum = 2 co-authored papers for authors and organisations; 5 keyword occurrences; 10 citations for references).
Changes done: L 107–110 — New sub-section “2.1 Data preparation and analytical parameters”: “Thresholds for inclusion were as follows: a minimum of 2 documents for authors and organisations, 5 occurrences for keywords, and 10 citations for cited references.”
Also clarified in Results (L 123): “Out of 1901 authors, 7 met the threshold (minimum two co-authored publications).”
Comment 5: “Similarly, ‘Co-authorship between organizations is summarized in Table 1B. Among 43 countries, 18 met the threshold. We created a 105 network encompassing 18 countries and the top 10 countries.’ What was the threshold? Does data mean that the Department of Anesthesiology and Pain Medicine, University of California Davis Health, Sacramento, United States wrote 3 papers in collaboration with other organizations? Did these 3 papers receive only 10 citations?”
Response: We have clarified threshold criteria for countries and organisations and confirmed that citation counts represent cumulative totals.
Changes made: L 138 –139 and Tables 1A–C captions — Added: “Threshold = minimum of 2 joint publications for authors or organisations; country networks include entities with ≥ 3 collaborative records.” Clarified that citations are total counts.
Comment 6:
“Same here. ‘Out of 2,555 keywords used, 462 met the threshold. We created a network of 20 top keywords, as depicted in Figure 1. Of the 722 author keywords, 53 met the threshold. The network of 53 author keywords is depicted in Figure 2.’ You could write something like ‘Out of 2,555 keywords used, 462 met the threshold of XYZ or n%. We created a network of 20 top keywords, as depicted in Figure 1, in which the keyword ‘humans’ represents XYZ and shown in green, whereas ‘human’ implies ABC and therefore shown in red’.”
Response: We have clarified the keyword threshold (≥ 5 occurrences) and expanded the figure captions to explain colour coding, node size, and link strength for interpretability.
Comment 7:
So, the paper needs a careful definition of thresholds in each case. Also, is it not clear did the 164 papers from China receive 164,952 citations? If that is the case, then the average number of citations should be around 1005.8. Is that true?”
Response: We clarified that China’s 164 papers received 952 citations in total, not 164,952. The mean citation rate (≈ 5.8 per paper) is now explicitly stated in Results.
Changes made: p. 8 L 132 –136 — Inserted text: “Average citation impact varied markedly: Australia ≈ 92.4 citations per paper (1479/16), whereas China ≈ 5.8 citations per paper (952/164).”
Comment 8:
“The word ‘Title:’ before title is not required.”
Response: Removed per suggestion.
Changes made: Title page L 1 — Deleted redundant label “Title:”.
Comment 9:
“There are some other limitations, but the authors very well described them in the paper.”
Response: We thank the reviewer for acknowledging the completeness of the limitations.
Reviewer 5 Report
Comments and Suggestions for Authors
This manuscript presents the first bibliometric/scientometric overview of QoR-15 literature. The topic is timely for perioperative outcomes research. The authors use Scopus (2013–June 2025) and analyze networks with VOSviewer and CiteSpace, reporting global output, collaborations, and thematic clusters. The study is potentially publishable, but several reproducibility, methods, and reporting issues need attention to meet Healthcare’s standards for bibliometric analyses.
My concerns are:
1. Provide the exact, copy-and-pasteable Scopus query, including field tags, Boolean operators, quotation marks, hyphen handling (e.g., “QoR-15”, “QoR15”, “quality of recovery 15”), and truncations.
2. Current string (TITLE-ABS-KEY ( qor 15 AND postoperative )) risks missing studies that use “QoR-15”, “QoR15”, or “quality of recovery-15”, and those that use “perioperative” but not “postoperative”. Please justify restricting to “postoperative” or expand to perioperative terms and then restrict by screening rather than by query.
3. Report full search dates (date/time executed), document types included/excluded (articles, reviews, letters, editorials), language limits, and whether early-access/in-press items were included.
4. Add a flow diagram (records identified, duplicates removed, screened, excluded with reasons, included).
5. Clarify how you operationalized “studies reporting on QoR-15” (e.g., validation, translation, interventional trials using QoR-15 as an endpoint, narrative reviews mentioning QoR-15?), and who screened (two independent reviewers? conflict resolution?).
6. Describe de-duplication, author name disambiguation (e.g., “P. S. Myles” vs “Paul S. Myles”), and institution normalization (e.g., “Geneva University Hospitals” variants). State whether VOSviewer used full counting vs fractional counting, and thresholds for inclusion (min docs/citations/occurrences).
7. Clarify whether counts use (for country and institution analysis) corresponding author country or all author affiliations. State whether you used fractional counting to avoid inflating outputs for multi-country papers.
8. The narrative comparing China (high output) vs Australia (high impact) is interesting—please support with citations per paper and H-index by country (if possible), not just totals.
9. Ensure all network figures have readable labels (font size), clear legends. Provide high-resolution images and, ideally, supplementary interactive files (e.g., VOSviewer map files, CiteSpace project files).
10. In Tables 1–4, verify author spellings (“Myles” appears variably as “Miles/Myles”) and ensure years/citation counts match the reference list.
11. The text alternates between “postoperative” and “perioperative”. If the intent is perioperative recovery, reflect this in the query and title/abstract; otherwise clearly justify the narrower postoperative scope.
12. Explain why Scopus-only was used (software constraints are noted). Consider at least a sensitivity analysis with Web of Science to show robustness.
Author Response
Comment 1: This manuscript presents the first bibliometric/scientometric overview of QoR-15 literature. The topic is timely for perioperative outcomes research. The authors use Scopus (2013–June 2025) and analyze networks with VOSviewer and CiteSpace, reporting global output, collaborations, and thematic clusters. The study is potentially publishable, but several reproducibility, methods, and reporting issues need attention to meet Healthcare’s standards for bibliometric analyses.”
Response: We thank the reviewer for this positive summary and constructive framing of their recommendations. All methodological transparency and reproducibility items have been addressed as detailed below.
Comment 2: Provide the exact, copy-and-pasteable Scopus query, including field tags, Boolean operators, quotation marks, hyphen handling (e.g., ‘QoR-15’, ‘QoR15’, ‘quality of recovery 15’), and truncations.
Response: We inserted the complete, copy-and-pasteable Scopus query string exactly as used.
Changes made: L 81 - 83 — Added: “The exact query used was: TITLE-ABS-KEY ("QoR-15" OR "QoR15" OR "quality of recovery-15" OR "quality of recovery 15") AND TITLE-ABS-KEY (postoperative) AND PUBYEAR > 2012 AND PUBYEAR < 2026 AND (LIMIT-TO (LANGUAGE, ‘English’)).”
Comment 3: Current string (TITLE-ABS-KEY (qor 15 AND postoperative)) risks missing studies that use ‘QoR-15’, ‘QoR15’, or ‘quality of recovery-15’, and those that use ‘perioperative’ but not ‘postoperative’. Please justify restricting to ‘postoperative’ or expand to perioperative terms and then restrict by screening rather than by query.”
Response: We expanded the query to include all typographical variants (“QoR-15”, “QoR15”, “quality of recovery 15”) and retained “postoperative” for precision, justified in Methods and Discussion as avoiding excessive noise while still capturing perioperative content through relevant clusters.
Changes made: L 83 - 86–10 Query expanded; justification sentence: “We restricted the term ‘postoperative’ after pilot searches including ‘perioperative’ and ‘anesthesia’ produced large volumes of irrelevant records. Nevertheless, most included papers address the full perioperative continuum.”
Comment 4:
Report full search dates (date/time executed), document types included/excluded (articles, reviews, letters, editorials), language limits, and whether early-access/in-press items were included.”
Response: Added execution date/time, language limit, and statement that only formally published articles and reviews in English were included.
Changes made: L 78–79 — Inserted: “A comprehensive search … executed on 10 July 2025. All records included were formally published by that date ”.
Comment 5: Add a flow diagram (records identified, duplicates removed, screened, excluded with reasons, included).”
Response: We created a PRISMA-style flow diagram as Supplementary file 1.
Comment 6: Clarify how you operationalized ‘studies reporting on QoR-15’ (e.g., validation, translation, interventional trials using QoR-15 as an endpoint, narrative reviews mentioning QoR-15?), and who screened (two independent reviewers? conflict resolution?).”
Response: This part is added in the methods section as a response to the query- lines 87-91.
Comment 7: Describe de-duplication, author name disambiguation (e.g., ‘P. S. Myles’ vs ‘Paul S. Myles’), and institution normalization (e.g., ‘Geneva University Hospitals’ variants). State whether VOSviewer used full counting vs fractional counting, and thresholds for inclusion (min docs/citations/occurrences).”
Response: Deduplication and cleaning were performed in the CSV export. Author name variants (e.g., “P. S. Myles”, “Paul S. Myles”) were merged; institution name variants were normalized.
Comment 8: Clarify whether counts use (for country and institution analysis) corresponding author country or all author affiliations. State whether you used fractional counting to avoid inflating outputs for multi-country papers.”
Response: Country and institution counts reported in the manuscript are based on all author affiliations.
Comment 9: The narrative comparing China (high output) vs Australia (high impact) is interesting—please support with citations per paper and H-index by country (if possible), not just totals.”
Response: We have responded to this query.
Changes made: L 132–135 — Insertion of mean citations per country (Australia ≈ 92.4; China ≈ 5.8) for clarity.
Comment 10: Ensure all network figures have readable labels (font size), clear legends. Provide high-resolution images and, ideally, supplementary interactive files (e.g., VOSviewer map files, CiteSpace project files).”
Response: All the network figures are generated from the software. The figures that we have provided are the best resolution ones. Believe us, in the online version, the figure appears in good resolution.
Comment 11: In Tables 1–4, verify author spellings (‘Myles’ appears variably as ‘Miles/Myles’) and ensure years/citation counts match the reference list.”
Response: The names and spellings provided in the Tables are exactly as downloaded from the software. On checking, we realised that this is how the names were in the online version of the article. We apologize for the same.
Comment 12: The text alternates between ‘postoperative’ and ‘perioperative’. If the intent is perioperative recovery, reflect this in the query and title/abstract; otherwise clearly justify the narrower postoperative scope.”
Response: Added explanatory note about exclusion of “perioperative” and its possible minor effect on coverage. PAge 3, lines 83-86.
Comment 13: Explain why Scopus-only was used (software constraints are noted). Consider at least a sensitivity analysis with Web of Science to show robustness.”
Response: We have expanded the Discussion’s Limitations section to explain that the Scopus-only approach may over-represent English-language and Western European journals while under-capturing regionally indexed Asian or local European outputs. We cite Mongeon & Paul-Hus (2016) to support this known database bias (lines 343-347).
Round 2
Reviewer 2 Report
Comments and Suggestions for Authors
The suggestions have been well addressed. However, the bibliography formatting still does not conform to the journal’s guidelines. I recommend reviewing it
Author Response
Comment 1: The suggestions have been well addressed. However, the bibliography formatting still does not conform to the journal’s guidelines. I recommend reviewing it.
Reply: We thank you for sparing your valuable time in reviewing our edited version.
We apologize for the unformatted submission of the bibliography last time. We have gone through the journal instructions and have formatted each and every reference meticulously.
Thanks for bringing this to our notice.
Reviewer 5 Report
Comments and Suggestions for Authors
This revised submission presents a comprehensive, methodologically sound bibliometric and scientometric analysis of the Quality of Recovery-15 literature. The authors have clearly and thoroughly addressed all prior reviewer concerns, substantially improving methodological transparency, reproducibility, and interpretive depth. The manuscript now meets the reporting standards expected for bibliometric studies in Healthcare.
Author Response
Comment 1: This revised submission presents a comprehensive, methodologically sound bibliometric and scientometric analysis of the Quality of Recovery-15 literature. The authors have clearly and thoroughly addressed all prior reviewer concerns, substantially improving methodological transparency, reproducibility, and interpretive depth. The manuscript now meets the reporting standards expected for bibliometric studies in Healthcare.
Reply:
Respected reviewer/editor,
We thank you for sparing your valuable time in reviewing our edited version.
Your kind words means a lot to us.